# How Collectivism and Virtual Idol Characteristics Influence Purchase Intentions: A Dual-Mediation Model of Parasocial Interaction and Flow Experience

**DOI:** 10.3390/bs15050582

**Published:** 2025-04-25

**Authors:** Yang Du, Wenjing Xu, Yinghua Piao, Ziyang Liu

**Affiliations:** Department of Global Business Graduate School, Kyonggi University, Suwon 16227, Republic of Korea; duyang@kyonggi.ac.kr (Y.D.); wjxuuu@kyonggi.ac.kr (W.X.); rndkpark@kyonggi.ac.kr (Y.P.)

**Keywords:** virtual idols, collectivism, parasocial interaction, flow experience, social identity theory, flow theory

## Abstract

With the rise of virtual idols in marketing, especially in collectivist cultures, their impact on consumer behavior warrants further exploration. This study applies social identity theory, flow theory, and the SOR model to examine how collectivism and virtual idol characteristics (external characteristics, content features, and homophily) influence Chinese consumers’ purchase intentions through parasocial interaction (PSI) and flow experience. A survey of 496 respondents, analyzed via structural equation modeling (SEM), shows that collectivism and virtual idol characteristics positively affect PSI, with homophily having the strongest impact. PSI enhances flow experience, and both PSI and flow experience drive purchase intention. PSI and flow experience serve as dual mediators in the model. This study advances research by empirically validating collectivism’s role in PSI, differentiating virtual idol characteristics, and modeling dual mediation. The key contributions of this study are as follows: (1) treating culture as an independent variable to empirically examine its impact on psychological mechanisms, and (2) deconstructing virtual idol characteristics into three dimensions—external, content, and homophily—to reveal their distinct influence on consumer psychology. Findings offer strategic insights for brands, recommending a dual-track approach integrating cultural adaptation and feature design to enhance consumer engagement and purchasing behavior.

## 1. Introduction

With the advancement of information technology, an increasing number of organizations are leveraging idols to promote their products and services on social media ([58]). Celebrities such as singers, actors, athletes, and idol groups are often regarded as credible endorsers due to their influence and ability to attract large audiences ([82]). Establishing a connection between consumers and brands has become a key strategy for enhancing market differentiation, particularly in online promotional settings ([6]). However, as celebrity endorsements have proliferated, concerns have emerged regarding the unethical behavior of some endorsers, leading to increasing skepticism among consumers ([46]).

Against this backdrop, virtual idols have begun to shift from niche subcultural icons to a more mainstream presence, especially among Generation Z ([80]). Compared to human celebrities, virtual idols offer brands greater control over image, content production, and reputational risk, making them an attractive option in strategic marketing ([65]). Their digital nature enables enhanced cost-efficiency and technological appeal, which further contributes to growing consumer engagement ([2]). Virtual idols are playing an increasingly prominent role in cultivating fan communities through activities such as virtual concerts and interactive engagement on social media platforms. The growth of their follower bases has been accompanied by demonstrable commercial outcomes, highlighting their strategic relevance in brand marketing ([71]). As their adoption continues to expand, virtual idols are becoming an integrated and adaptable element within brand communication practices.

The deep integration of virtual idols with ACG (Anime, Comic, and Game) culture has attracted a large number of young consumers, making them a vital medium for brands to connect with digital natives. While previous studies have identified common attributes of virtual idols, these attributes are largely derived from the existing literature and may not fully capture the emergence of new virtual idol forms or the evolving perceptions of users ([88]). Meanwhile, research on AI influencers has developed attribute-based measurement scales and examined their direct effects on consumer acceptance ([18]). Although recent studies have advanced the understanding of virtual idol endorsements, much of the existing research continues to focus on their direct effects ([2]; [11]; [53]), with relatively little attention paid to the underlying psychological mechanisms or the roles of cultural context and idol-specific characteristics in shaping consumer behavior.

In the East Asian cultural context, as studied by [25] ([25]), countries in this region exhibit a high degree of collectivism, which influences consumer social behavior and purchasing decisions. Over time, improvements in sociopolitical conditions up to 2024 have not led to increased individualism ([44]). In China specifically, consumers are more inclined to focus on others, integrate into social groups, and maintain harmonious interdependence ([57]), which sharply contrasts with the individualistic tendencies of Western markets ([64]). This cultural divergence presents a unique research opportunity to explore the psychological pathways and culture-driven mechanisms underlying the effects of virtual idol endorsements.

The current literature on virtual idol endorsement mechanisms still suffers from an oversimplification of characteristics. Mainstream studies, largely conducted within Western individualistic contexts, emphasize the personalized narratives of virtual idols ([92]) while overlooking the distinct demand for “group identity symbols” among collectivist consumers. For instance, A-SOUL resonates with fans through Chinese-style character designs, reinforcing a sense of identity. Some studies have confirmed that virtual idols can enhance purchase intentions among Chinese consumers ([33]), yet they fail to clarify whether collectivist traits drive a sense of group belonging toward virtual idols. Moreover, prior research has not differentiated the varying impact pathways of virtual idols’ external characteristics, content characteristics, and homophily. While some studies have examined consumer choices regarding virtual idols endorsing different product categories ([98]), they have not categorized the specific influences of virtual idol characteristics on consumer behavior. Additionally, research has found that consumer emotions affect their willingness to pay for virtual idols ([53]), but these studies have yet to address how the intrinsic characteristics of virtual idols influence consumer psychology.

Existing research has extensively explored the influence mechanisms of parasocial interaction (PSI) on consumers’ purchase intentions. Studies have shown that PSI between social media opinion leaders and users can significantly enhance purchase intention ([77]). Particularly in live-streaming scenarios, PSI between streamers and audiences not only serves as a key driver of purchasing decisions but also mediates the relationship between persuasive strategies and consumer behavior ([75]). However, current research primarily focuses on endorsements by real-life idols ([17]) and live-streaming sales models ([55]), while the role of PSI in virtual idol marketing remains underexplored.

Meanwhile, flow experience, an emerging perspective in consumer behavior research, has been validated across multiple domains. In live-streaming marketing, flow experience is significantly positively correlated with purchase intention and exhibits a mediating effect ([91]). Additionally, studies on digital products ([61]) and live-streaming e-commerce ([54]) have confirmed that flow experience enhances purchase intention through a heightened sense of immersion, also serving as a mediator. However, existing research presents two notable limitations: first, the mechanisms of flow experience in virtual idol marketing have yet to be systematically analyzed. second, most studies have examined either the affective pathway (PSI) or the cognitive pathway (flow experience) in isolation, failing to integrate the synergistic effects of both.

Therefore, this study integrates social identity theory and flow theory to construct a pathway model addressing the following questions:

1. How does collectivism directly influence consumers’ PSI? 2. What are the differential effects of virtual idols’ external characteristics, content characteristics, and homophily on PSI? 3. How does PSI influence flow experience, and how does flow experience mediate the effects of PSI on purchase intention? 4. More importantly, can collectivism and virtual idol characteristics influence purchase intention through PSI and flow experience?

Based on the above theoretical framework and research questions, this study adopts the SOR model, treating collectivism and virtual idol characteristics as independent variables. It explores their differentiated impact pathways on consumers’ psychological mechanisms (PSI and flow experience) and examines how these psychological responses (PSI and flow experience) influence behavioral outcomes (purchase intention). Through this approach, the study systematically analyzes how virtual idol endorsements shape consumer behavior within China’s collectivist cultural context.

The innovations of this study are as follows: 1. It overcomes the limitation of consumer behavior research that simplifies culture as a moderating variable ([20]; [41]), by treating culture as an independent variable and empirically testing its independent influence on psychological mechanisms. 2. It deconstructs the characteristics of virtual idols into external characteristics (visual experience), content characteristics (interactive experience), and homophily (resonance), and through the ”external–content–homophily” three-dimensional classification, reveals the differentiated influence pathways of these characteristics on consumer psychology.

The theoretical significance of this study lies in expanding the application of the SOR model in virtual idol endorsements and cross-cultural contexts. On the practical level, it demonstrates that brands in the Chinese market need to adopt a dual-track approach—actively adapting to collectivist values to integrate virtual idols into local cultural symbols, while also carefully designing feature combinations to achieve the synergistic effects of psychological mechanisms.

## 2. Literature Review and Hypotheses Development

### 2.1. Virtual Idol

Virtual idols are digital characters created using 3D modeling, artificial intelligence, and motion capture technology ([78]). Their essence lies in simulating the interaction abilities and personalized characteristics of human idols through algorithms ([7]). With technological advancements, virtual idols have gradually transitioned from subcultures to mainstream commercial applications ([51]) and have become emerging marketing tools in the luxury goods, fashion, and electronics sectors ([16]). For instance, Dior employed the virtual idol Noonoouri as a brand ambassador, while the Chinese virtual group A-SOUL achieved sales exceeding 100 million yuan in a single livestream event. The substantial purchasing power of Generation Z has driven brands to incorporate virtual idols as brand ambassadors in their social media marketing campaigns ([92]), particularly in East Asia. As a result, virtual idols who are highly admired and beloved by fans—such as Hatsune Miku, Noa, Angie, A-SOUL, and Ling—have emerged as preferred digital brand endorsers. Examples of virtual idols are presented in Table 1. This study deconstructs the characteristics of virtual idols into three dimensions: external characteristics, content characteristics, and homophily, and analyzes the mechanisms behind their influence.

External characteristics refer to the visual design elements of a virtual idol, including facial features, clothing style, and dynamic expressions. Research has shown that highly attractive external features can rapidly capture consumer attention ([70]) and establish lasting impressions ([29]). For example, luxury brands tend to favor virtual idols with minimalist styles, ensuring that their external characteristics align closely with the brand’s tone ([71]).

Content characteristics encompass the narrative style, forms of interaction, and value output of virtual idols. High-quality content must possess originality, hedonic value, and professionalism ([3]; [22]). Such content significantly enhances consumer trust and purchase intention by triggering emotional resonance and cognitive immersion ([90]).

Homophily refers to the perceived similarity between virtual idols and consumers in terms of values, cultural symbols, or social identity ([67]). Research indicates that when the image of a virtual idol aligns with the user’s ideal self or group affiliation needs, the endorsement effect is significantly enhanced ([37]).

### 2.2. Social Identity Theory

Social identity theory posits that the core of individual identity lies in one’s membership within a group ([81]). The concept of social identity can be divided into two subdimensions: cognitive identification and affective identification ([28]; [39]). Cognitive identification refers to the process of self-categorization as a member of a specific group ([39]), manifesting as the perceived overlap between group identity and self-concept ([72]). For example, fans of virtual idols engage in self-categorization by wearing community insignias. Affective identification describes an individual’s emotional attachment to and positive evaluation of their group membership ([39]), which is reflected in loyalty to the group and the need for belonging ([97]).

This dual identification mechanism drives individuals to regulate their behavior according to group norms ([4]). In the context of virtual idols, collectivist cultures amplify the effects of social identity by reinforcing group primacy ([64]; [86]). A typical example of group norm-driven behavior is the “boost popularity” phenomenon among Chinese virtual idol supporters—fan communities collectively boost idol rankings and control online discourse to maintain the idol’s popularity. Individual participants not only gain a sense of group belonging but also experience emotional gratification from contributing to the collective effort.

### 2.3. Flow Theory

Flow experience, a concept introduced by [15] ([15]), refers to a state of deep immersion and intense enjoyment in an activity, during which individuals tend to ignore external stimuli. In online environments, flow is characterized as a cognitive state marked by seamless responsiveness, interactivity, intrinsic enjoyment, self-reinforcement, and a diminished sense of self-awareness ([62]). Thus, when fans watch virtual idol videos, they may enter a flow state due to the high level of interactivity, becoming fully immersed in the content experience ([10]).

Individuals in a flow state not only experience heightened enjoyment and a sense of behavioral control ([47]) but also exhibit increased receptivity to product information due to their sense of immersion, ultimately leading to enhanced purchase intentions ([36]; [52]).

### 2.4. Parasocial Interaction (PSI)

Parasocial interaction (PSI) was originally introduced in the field of communication studies to describe the illusion of a one-sided emotional bond between individuals and media figures such as television hosts, celebrities, or fictional characters ([26]). Although these relationships lack genuine reciprocity, audiences may perceive media figures as close acquaintances, engaging in imagined conversations and experiencing a sense of familiarity and trust ([27]; [66]). Due to its emphasis on emotional intimacy and psychological involvement, PSI has become a widely adopted framework for understanding consumer behavior ([45]). With the rapid development of digital media, the scope of PSI has significantly expanded beyond traditional broadcasting contexts. Online platforms now offer a diverse range of entities—such as influencers, livestreamers, and virtual characters—with whom users can form parasocial bonds ([38]). Features of social media, including direct messaging, comment sections, and real-time updates, enhance the perception of interpersonal engagement and deepen the emotional connection ([40]; [76]). The increasing popularity of short-form content has also altered the temporal structure of PSI, often leading to more frequent yet shorter interactions ([75]). In addition, media figures frequently engage in self-disclosure on social platforms, sharing personal and professional details that foster a perceived sense of intimacy similar to real-life friendships ([50]). In the context of virtual idols, PSI has become especially relevant. Even in the absence of real human agency, audiences can form meaningful emotional bonds with virtual figures through visual realism, consistent character narratives, and algorithmically mediated interactions ([56]; [79]). In immersive environments such as the metaverse, virtual influencers are increasingly being used in place of traditional celebrity endorsers to target digitally native audiences with innovative and culturally adaptive marketing strategies ([89]). Although these interactions are entirely computer-mediated, users often perceive them as authentic and responsive, which further strengthens parasocial ties. Sustained engagement through PSI may eventually evolve into parasocial relationships (PSRs), which have been associated with increased brand loyalty and purchase intentions ([96]). Moreover, PSI serves as a key affective driver of user engagement, enhancing emotional resonance and behavioral participation ([87]).

Figure 1 is presented following the theoretical background to illustrate the hypothesized relationships among the study variables.

### 2.5. Hypotheses Development

#### 2.5.1. Collectivism and PSI

Collectivist culture constructs consumers’ self-concepts through in-group identity symbols, suggesting that collectivism serves as a critical driving force in the formation of PSI ([5]). In cross-cultural comparisons, consumers with high collectivism are more inclined to establish PSI with virtual idols compared to those with high individualism ([48]), thereby confirming the explanatory power of collectivism as a core dimension in cultural classification ([73]). Specifically, collectivist consumers exhibit a significant preference for anthropomorphized products ([5]), and their decision-making processes are heavily reliant on group consensus—individuals actively internalize the expectations, preferences, and opinions of group members in order to attain social identity ([9]). Although the link between collectivism and PSI is well established, its implications in the context of virtual idols remain underexplored. The anthropomorphic features of virtual idols may influence how cultural values are experienced and expressed in these interactions. Therefore, a hypothesis of the study is as follows:

**H1:** 
*Collectivism has a positive impact on PSI.*


#### 2.5.2. Virtual Idol Characteristics and PSI

In this study, the features of virtual idols are deconstructed into three independent dimensions—external characteristics, content characteristics, and homophily—which jointly drive the formation of consumers’ PSI. The underlying mechanisms are as follows:

First, external characteristics trigger PSI through visual symbols. Studies have shown that high-fidelity external design can evoke positive emotions ([33]) and significantly enhance purchase intentions by reinforcing brand perception ([12]; [74]). Brands leverage the high visibility of virtual idols to capture consumer attention and to communicate more positive attitudes toward the brand, thereby promoting product sales ([60]). Although visual features have been widely acknowledged as influential, their specific contribution to the development of PSI has received limited theoretical attention. Therefore, a hypothesis of the study is the following:

**H2:** 
*Virtual idols’ external characteristics has a positive impact on PSI.*


Second, content characteristics sustain PSI through interactive experience. The creative content and high interactivity design of virtual idols construct an immersive participation environment. Firstly, creative content, by providing hedonic value, extends user immersion time and fosters PSI between consumers and virtual idols ([92]). Furthermore, highly interactive content simulates real social scenarios ([43]; [45]), leading fans to regard frequently interacting virtual idols as “intimate friends” ([23]). For instance, fans engaging in real-time interactions via bullet chats develop a sense of friendship-like belonging and loyalty. Therefore, a hypothesis of the study is the following:

**H3:** 
*Virtual idols’ content characteristics has a positive impact on PSI.*


Third, homophily refers to the perceived similarity between consumers and virtual idols regarding external appearance, values, or interests ([14]). Its influence is reflected in two dimensions. First, when a virtual idol’s external characteristics (e.g., facial contours and clothing styles) match a consumer’s ideal self-image or group aesthetic standards, a stronger sense of social attractiveness is triggered ([77]). For example, virtual idols designed for teenagers often adopt an anime style (e.g., large eyes and trendy outfits). Second, when the values conveyed by virtual idols through their words and actions align with those of consumers, psychological closeness is significantly enhanced ([95]). In this study, both appearance-based and ideational homophily were incorporated into the measurement design to examine how virtual idols influence PSI. Therefore, a hypothesis of the study is the following:

**H4:** 
*Virtual idols’ homophily has a positive impact on PSI.*


#### 2.5.3. The Impact of PSI and Flow Experience on Purchase Intentions

Based on existing research, it is evident that PSI and flow experience play crucial roles in influencing consumers’ purchase intentions. Moreover, PSI and flow experience serve as dual mediators in consumer behavior by transmitting the effects of collectivism and virtual idol features on purchase intentions through affective and cognitive pathways. Specifically:

PSI influences consumer decision-making through emotional attachment and identity formation. Research has shown that PSI between online celebrities and users positively promotes purchase intentions ([77]). This effect is particularly pronounced in virtual idol contexts, where the appeal of virtual idols indirectly enhances consumer conversion rates by reinforcing PSI ([13]). Empirical data from the Chinese market further validate that the level of PSI between audiences and virtual characters is significantly and positively correlated with purchase intentions ([35]). Collectivist culture amplifies this effect through group identity symbols; consumers, guided by cultural values, perceive their purchase behaviors as expressions of group identity ([93]). For instance, the premium paid for sustainable apparel implicitly reflects an identification with group values ([42]; [97]).

PSI not only directly affects purchase intentions but also influences consumers’ flow experience. Research indicates that PSI impacts flow states through the following mechanisms: by establishing emotional attachment, PSI extends user engagement, thereby providing a sustained basis for triggering flow experience ([29]); additionally, PSI stimulates consumers’ resonance with virtual idols, leading them to focus their attention intensely on interactive content ([32]). This concentrated engagement reduces the perception of external distractions, thus facilitating the onset of a flow state ([91]).

Flow experience, in turn, affects purchase intentions through cognitive focus and a sense of behavioral control. Interactions conducted via computer interfaces can induce a state of focused attention, culminating in flow experience ([32]). Human–machine interactions significantly enhance the flow experience, while social interactions further promote immersion ([30]). Studies have demonstrated that flow experience plays a mediating role in both livestreaming e-commerce ([91]) and digital consumption scenarios ([61]), with its core logic being that immersion reduces decision-making resistance and consequently strengthens purchase motivation ([29]).

Moreover, PSI and flow experience do not operate in isolation; rather, they form a chain mediation through an affective–cognitive synergy. For example, the anthropomorphic features of virtual idols (such as culturally symbolic design) activate a sense of group belonging via PSI and simultaneously reinforce immersive participation through flow experience, ultimately driving purchase behavior. Although this mechanism has been partially validated in studies on real-life idols ([17]) and livestreaming sales ([55]), its cultural adaptability in virtual idol marketing warrants further investigation. Although PSI and flow have been widely examined as individual predictors of consumer behavior, their combined mediating role remains underexplored, particularly in culturally embedded contexts such as virtual idol marketing. Based on the foregoing discussion, hypotheses of the study are as follows:

**H5:** 
*PSI has a positive impact on flow experience.*


**H6:** 
*PSI has a positive impact on purchase intentions.*


**H7:** 
*Flow experience has a positive impact on purchase intentions.*


**H8:** 
*Collectivism indirectly influence purchase intentions through the dual mediation of PSI and flow experience.*


**H9:** 
*Virtual idols’ external characteristics indirectly influence purchase intentions through the dual mediation of PSI and flow experience.*


**H10:** 
*Virtual idols’ content characteristics indirectly influence purchase intentions through the dual mediation of PSI and flow experience.*


**H11:** 
*Virtual idols’ homophily indirectly influences purchase intentions through the dual mediation of PSI and flow experience.*


## 3. Research Methodology

### 3.1. Measurement

To ensure the content validity of the questionnaire, all measurement items were adapted from well-established scales in prior research. Items measuring collectivism were adapted from [85] ([85]). The characteristics of virtual idols were measured based on scales developed by [59] ([59]) and [63] ([63]). Parasocial interaction items were adapted from [69] ([69]). The concept of flow experience was assessed using the measurement approach proposed by [68] ([68]), while purchase intention was measured using items from [49] ([49]). All constructs were measured using a 7-point Likert scale (1 = strongly disagree, 7 = strongly agree) following the format suggested by [84] ([84]). A detailed list of the measurement items is provided in Appendix A and Appendix B.

### 3.2. Data Collection

This study utilized www.wjx.cn to design the questionnaire, which was then distributed online via the WeChat social networking application. As the most widely used social media platform in China, WeChat has approximately 1.2 billion active users. The survey was conducted under strict anonymity, and respondents were provided with a monetary incentive of one to three RMB for completing the questionnaire. Considering that virtual idols are an emerging internet phenomenon, a screening question was included in the questionnaire to ensure data validity. Respondents who were unfamiliar with virtual idols were not allowed to proceed, thereby preventing invalid responses from affecting the results. Ultimately, 496 valid responses were collected, which were subsequently used for reliability and validity testing as well as regression analysis.

To provide a clear demographic background for the study, a descriptive statistical analysis was conducted on the sample’s characteristics in terms of gender, age, occupation, and income. The results in Table 2 show that female respondents accounted for 55.6%, slightly higher than males at 44.4%. In terms of age distribution, the largest proportion of respondents fell within the 26–34 age group, comprising 47.0% of the sample, followed by the 35–42 age group at 26.4%, indicating that the sample is primarily composed of young and middle-aged individuals. Regarding occupation, corporate employees made up the largest proportion at 57.3%, significantly higher than students at 17.7%, self-employed individuals and entrepreneurs at 16.7%, and government employees at 8.3%. In terms of income, the highest proportion of respondents, 37.5%, reported an income between USD 801 and USD 1000, followed by those earning less than USD 800 at 24.0%. This indicates that the sample primarily consists of middle-income groups, while only 9.4% of respondents reported an income exceeding USD 1600. These demographic characteristics reflect both the diversity and concentration trends within the sample, providing a solid foundation for further analysis.

## 4. Results

### 4.1. Reliability and Validity Test

Statistical analysis was conducted using SPSS 23, and the overall KMO was found to be 0.812, exceeding the recommended threshold of 0.8, confirming that the dataset was suitable for further analysis. KMO test are key to evaluating data adequacy and construct validity, ensuring the reliability and rigor of factor analysis. The PLS algorithm in Smart-PLS 4.0 was applied, with the maximum number of iterations set to the default value of 300. The results indicated that all factor loadings in the sample met or exceeded the standard threshold of 0.7.

In this study, all constructs exhibited Cronbach’s α values exceeding the standard threshold of 0.7, demonstrating strong internal consistency. Additionally, the AVE values for all constructs were above the recommended threshold of 0.5, further validating the model’s internal consistency. Another measure of reliability, CR was also assessed. The results showed that all CR values significantly exceeded 0.7, indicating strong reliability. The results are shown in Table 3.

Discriminant validity was confirmed using the Fornell and Larcker method, which evaluates discriminant validity by examining whether the square root of the AVE for each construct exceeds the correlation coefficients between constructs ([19]). Based on this criterion, discriminant validity was successfully established. The results presented in Table 4 show that the smallest square root of AVE in the sample was 0.829, which was significantly greater than the highest correlation coefficient of 0.278 ([94]).

Discriminant validity was further verified using the Heterotrait–Monotrait (HTMT) ratio of correlations, which should be significantly lower than 0.85 ([21]). The results in Table 5 confirm that the data exhibit sufficient discriminant validity.

### 4.2. Multicollinearity and Explanatory Power Test

First, the variance inflation factor (VIF) was used to assess multicollinearity. The results indicated that all VIF values for the structural variables were well below 3.3, suggesting that multicollinearity was not a serious issue. The model’s explanatory power was also tested. The results showed that the R^2^ values for PSI (R^2^ = 0.154), flow experience (R^2^ = 0.039), and purchase intention (R^2^ = 0.093) were all above the reference threshold, demonstrating that the model provides a strong explanatory ability for these variables ([24]). Additionally, as shown in Table 6, the F^2^ test results exceeded the reference standard, indicating that the independent variables had a sufficient explanatory effect on the dependent variables ([34]).

Furthermore, the Q^2^ values for PSI (Q^2^ = 0.107), flow experience (Q^2^ = 0.028), and purchase intention (Q^2^ = 0.064) were all greater than zero, confirming the model’s predictive relevance ([21]). Finally, the model fit was assessed, yielding an SRMR value of 0.047, which met the recommended threshold, indicating a good model fit ([31]) and supporting its suitability for further analysis.

### 4.3. Hypothesis Testing

Path analysis was conducted using Smart-PLS 4.0, with 5000 resampling bootstraps applied. The specific results are presented in Table 7 (the structural model diagram is shown in Figure 2). The analysis revealed that all *t*-values were greater than 1.96, and all *p*-values were below 0.01, strongly supporting the proposed hypotheses. All paths met the hypothesis testing criteria, confirming the validity of the research hypotheses.

Additionally, path coefficients indicated that homophily was the most significant predictor of PSI among the virtual idol characteristics. As shown in Table 8, examining whether PSI and flow experience serve as dual mediators between collectivism, virtual idol characteristics, and consumer purchase intention is crucial. The results demonstrated significant mediation effects for H8 (β = 0.008, *t* = 2.526), H9 (β = 0.007, *t* = 2.363), H10 (β = 0.007, *t* = 2.551), and H11 (β = 0.008, *t* = 2.647). The independent variables exert both direct effects on purchase intention and indirect effects through a sequential mediation process. This process involves two consecutive psychological mechanisms: the development of PSI, followed by the elicitation of immersive flow experience during user engagement, which together contribute to the enhancement of consumers’ purchase intention.

Furthermore, the VAF values confirmed that H8–H11 exhibited partial mediation effects ([23]).

## 5. Discussion and Implications

### 5.1. Discussion

Taking Chinese consumers as a sample, this study constructs a theoretical pathway based on structural equation modeling to explore how collectivism and virtual idol characteristics directly or indirectly influence PSI, flow experience, and purchase intention. Overall, the empirical results support all proposed hypotheses and confirm the dual mediating roles of PSI and flow experience in transmitting the effects of collectivism and virtual idol characteristics (i.e., external characteristics, content characteristics, and homophily) on purchase intention ([16]; [33]; [35]).

Firstly, regarding H1, the data indicate that collectivism significantly and positively influences parasocial interaction (PSI) between consumers and virtual idols ([5]; [49]). In addition, the findings corroborate that individuals tend to exhibit higher levels of PSI when they internalize group expectations ([9]). The results further suggest that individuals with a high degree of collectivist values are more likely to develop a sense of fan club membership and closely monitor the daily activities of virtual idols, thereby extending previous explanations of group belonging’s influence on PSI formation ([73]). Survey responses also indicate that respondents who prioritize team harmony tend to exhibit higher engagement when viewing virtual idol programs, which may serve as a precursor to subsequent purchase decisions.

This cultural orientation is also reflected in individuals’ attachment patterns. Prior research suggests that individuals from collectivist cultures generally exhibit stronger attachment tendencies, while those from individualistic cultures are more likely to emphasize emotional independence and relational autonomy ([1]). According to attachment theory, individuals with higher attachment tendencies are more likely to experience emotional closeness, dependence, and sustained attention toward their attachment targets ([83]). In digitally mediated environments, such attachment tendencies may promote the formation of parasocial interactions with virtual idols ([38]), but also enhance the depth of immersive experiences during these interactions. Accordingly, cultural values may shape the intensity of emotional bonding with virtual idols by influencing users’ attachment orientations, thereby offering a meaningful lens for understanding cross-cultural variation in PSI.

Secondly, regarding H2, H3, and H4, the study confirms that external characteristics, content characteristics, and homophily each exert a positive influence on PSI. The results indicate that external characteristics, conveyed through visual symbols, are significantly associated with the rapid capture of consumer attention; creative and interactive content characteristics are related to prolonged consumer engagement and the evocation of emotional resonance; and a higher degree of homophily is associated with greater consumer identification and attachment. Comparative analysis reveals that homophily exhibits the strongest positive association with PSI relative to external and content characteristics. Respondents generally endorsed statements suggesting that the virtual idol is close to their age, resembles them, and shares their way of thinking. This pattern suggests that perceived consistency in appearance, identity, and values between consumers and virtual idols is associated with enhanced feelings of belonging and emotional connection, which in turn relate to higher levels of PSI ([8]; [95]). In the context of Chinese collectivist culture, where group belonging and identity alignment are emphasized ([57]), homophily appears to be particularly salient. This finding extends the existing literature by suggesting that, for China’s digitally native consumers, perceived resonance may be more influential than appearance or content alone in driving parasocial interaction. In forming emotional bonds with virtual idols, consumers prioritize psychological and identity-based alignment over surface-level aesthetic appeal. This pattern may reflect a broader shift in personalized media environments, where virtual idols are increasingly perceived not merely as visual attractions, but as mirrors of personal identity.

Thirdly, regarding H5, H6, and H7, the results indicate that PSI is significantly and positively associated with flow experience, and that both PSI and flow experience are directly related to purchase intention. Specifically, higher levels of PSI are associated with greater immersion, which in turn is directly related to purchasing behavior ([29]; [30]). Furthermore, the data support the notion that flow experience functions as a mediator in consumer decision-making ([91]). Survey results suggest that flow is primarily triggered by consumers’ feelings of being needed and a sense of belonging during interaction. This indicates that PSI facilitates flow through emotional connection rather than task engagement. The findings also emphasize the central role of affective involvement in immersive experience, highlighting the importance of designing for emotional resonance rather than functional complexity in virtual idol interactions.

Finally, regarding H8, H9, H10, and H11, the findings indicate that, within the dual-mediation pathway, collectivist values significantly enhance consumers’ sense of belonging during interaction. In collectivist contexts, engagement with virtual idols is not solely driven by content preference or aesthetic appeal, but is shaped by a desire to extend social affiliation. This social orientation reinforces PSI and deepens immersion. Furthermore, the survey results show that external appearance, content features, and perceived homophily each exert significant positive effects on PSI, confirming that virtual idols operate as multidimensional symbolic constructs in facilitating emotional connection. These findings suggest that PSI arises not from a single perceptual cue, but from the combined influence of visual appeal, interactivity, and identity-based resonance. Accordingly, virtual idol design should adopt a holistic approach that integrates cognitive and emotional dimensions, rather than focusing narrowly on appearance or content, to foster sustained user immersion and engagement.

### 5.2. Implications for Practice

(1)Fostering Community

Collectivist cultures emphasize group belonging and social identity, both of which are closely tied to parasocial interaction (PSI). Brands should develop community-based engagement strategies—such as establishing official fan communities on platforms like WeChat or Douban and organizing Q&A sessions or creative contests—to strengthen users’ collective identification with virtual idols.

(2)Designing Virtual Idols for Perceived Homophily

The findings highlight homophily as a key driver of PSI. When designing virtual idols, brands should prioritize alignment with users’ values, aesthetics, and lifestyles. For example, virtual ambassadors like Ling resonate with Generation Z consumers through their tech-savvy, fashion-forward personas that reflect emerging cultural preferences.

(3)Creating Immersive Experiences through Interactive Technologies

Flow experiences triggered by PSI significantly enhance purchase intention. Brands can utilize technologies such as virtual reality (VR) and augmented reality (AR) to create immersive scenarios—for instance, holographic concerts, AR filters, or virtual meet-and-greets—to deepen user engagement and emotional involvement.

(4)Implementing a Multichannel, Data-Informed Strategy

An integrated online–offline approach can expand reach and increase user stickiness. Brands may showcase virtual idol personalities through short-form video platforms, while reinforcing real-world connection via fan festivals or pop-up events. Engagement data, including user comments and participation metrics, should be leveraged to refine content strategies. For example, KFC’s collaboration with A-SOUL incorporated fan feedback to tailor campaign content and enhance relevance.

By integrating cultural alignment, homophily-based design, immersive technology, and data-driven multichannel strategies, brands can strengthen users’ emotional bonds and immersive experiences with virtual idols, ultimately transforming psychological engagement into behavioral outcomes.

### 5.3. Theoretical Implications

First, this study incorporates social identity theory and flow theory into the context of virtual idol marketing and develops a dual-mediation model featuring PSI and flow experience. The model demonstrates the emotional and cognitive mechanisms through which consumers’ purchase intentions are shaped. In contrast with previous studies that tend to examine PSI and flow separately, this study provides empirical support for their combined and sequential effects. It expands our understanding of how consumers form psychological connections with non-human agents, and how such engagement drives behavioral responses in immersive digital environments.

Second, the study moves beyond the conventional treatment of culture as a moderating variable by positioning collectivism as a core independent variable within the PSI–flow–purchase intention framework. It highlights the active role of cultural values in triggering psychological engagement. The findings show that collectivist orientations enhance both parasocial bonding and immersive involvement with virtual idols, which in turn promote purchase intention. These results, drawn from a Chinese sample, broaden the theoretical scope of social identity theory and flow theory and provide a foundation for future research on how cultural values activate emotional and cognitive responses in virtual consumption settings.

### 5.4. Research Limitations and Future Research Suggestions

This study has several limitations that offer opportunities for future research.

First, the use of cross-sectional data limits the ability to capture changes in consumer behavior over time. Given the rapid evolution of virtual idol technologies and media environments, future studies may adopt longitudinal or experimental designs to examine the dynamic effects of virtual idol marketing.

Second, although the model is grounded in social identity theory and flow theory, future research could incorporate alternative theoretical perspectives—such as attachment theory or uses and gratifications theory—to further explore consumer interaction with virtual agents.

Third, this study does not consider potential moderating variables such as gender, digital literacy, or prior experience with virtual content. Future research could conduct subgroup analyses to explore heterogeneity in consumer responses across demographic segments.

Finally, as the sample was limited to Chinese respondents, the generalizability of the findings may be constrained. Future studies should include more diverse samples and consider cross-cultural comparisons. Moreover, attention should be paid to the potential for social desirability bias in self-reported data, in order to improve external validity.

## Figures and Tables

**Figure 1 behavsci-15-00582-f001:**
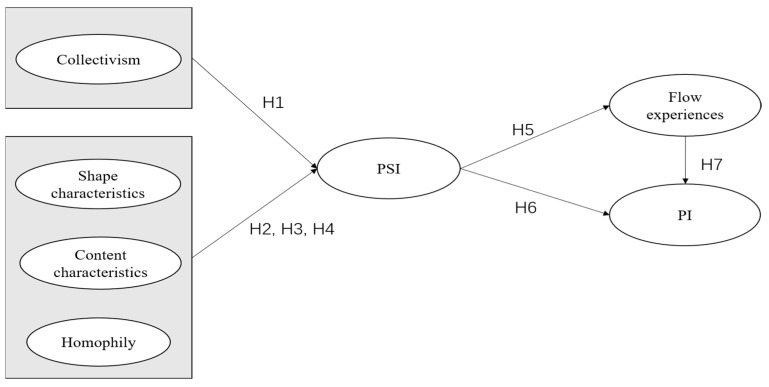
Research model.

**Figure 2 behavsci-15-00582-f002:**
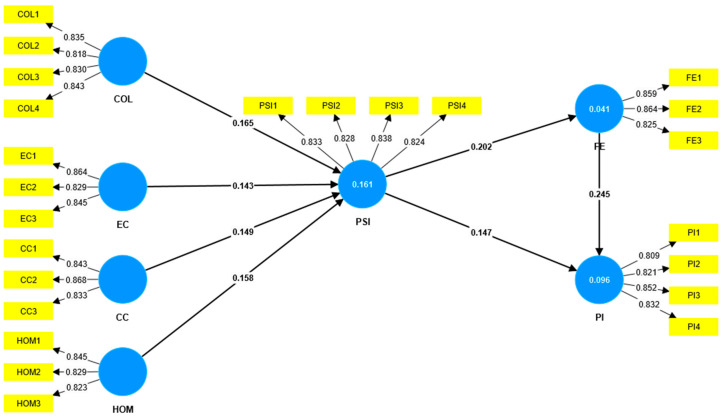
Structural model for this study.

**Table 1 behavsci-15-00582-t001:** Examples of virtual idol endorsements.

Virtual Idol	Marketing Case
Hatsune Miku	Toyota, BMW, Sony, Google, Rex, etc.
Angie	Yadi, OPPO, Chery, etc.
Noah	Shiseido, Haier, etc.
Ling	Tesla, Bulgari, Gucci, Lancôme, Avon, etc.
A-SOUL	L’Oreal Man, KFC, Asus, Keep, etc.

**Table 2 behavsci-15-00582-t002:** Demographic information.

Particulars	Description	Values	%
Gender	Male	220	44.4
	Female	276	55.6
Age (years)	18–25	88	17.7
	26–34	233	47.0
	35–42	131	26.4
	Older than 42	44	8.9
Occupation	Students	88	17.7
	Company employees	284	57.3
	Government employees	41	8.3
	Self-employment and entrepreneurs	83	16.7
Income (monthly)	Less than USD 800	119	24.0
	USD 801–1000	186	37.5
	USD 1000–1300	88	17.7
	USD 1301–1600	56	11.3
	More than USD 1600	47	9.4

**Table 3 behavsci-15-00582-t003:** Results of reliability and validity.

Construct	Loadings	VIF	Cronbach’s Alpha	CA	CR	AVE
Collectivism (COL)			0.851	0.853	0.900	0.692
COL1	0.835	1.902				
COL2	0.818	1.859				
COL3	0.830	1.941				
COL4	0.843	2.072				
External characteristics (EC)			0.802	0.807	0.883	0.716
EC1	0.864	1.740				
EC2	0.829	1.685				
EC3	0.845	1.742				
Content characteristics (CC)			0.805	0.817	0.885	0.719
CC1	0.843	1.828				
CC2	0.868	1.722				
CC3	0.833	1.695				
Homophily (HOM)			0.779	0.782	0.871	0.693
HOM1	0.845	1.642				
HOM2	0.829	1.550				
HOM3	0.823	1.654				
PSI			0.851	0.855	0.899	0.690
PSI1	0.809	1.920				
PSI2	0.821	1.971				
PSI3	0.852	1.852				
PSI4	0.832	1.951				
Flow experiences (FE)			0.808	0.817	0.886	0.722
FE1	0.859	1.827				
FE2	0.864	1.730				
FE3	0.825	1.724				
Purchase intention (PI)			0.848	0.851	0.898	0.687
PI1	0.809	1.794				
PI2	0.821	1.892				
PI3	0.852	2.019				
PI4	0.832	1.911				

**Table 4 behavsci-15-00582-t004:** Discriminant validity (Fornell–Larcker criterion).

Construct	CC	COL	FE	HOM	PI	PSI	SC
CC	0.848						
COL	0.254	0.832					
FE	0.219	0.243	0.850				
HOM	0.257	0.223	0.200	0.832			
PI	0.329	0.294	0.275	0.221	0.829		
PSI	0.269	0.268	0.202	0.261	0.197	0.831	
EC	0.260	0.205	0.252	0.194	0.278	0.246	0.846

**Table 5 behavsci-15-00582-t005:** Discriminant validity (HTMT).

Construct	CC	COL	FE	HOM	PI	PSI	SC
CC							
COL	0.303						
FE	0.265	0.289					
HOM	0.325	0.273	0.251				
PI	0.398	0.343	0.330	0.272			
PSI	0.322	0.311	0.236	0.317	0.231		
EC	0.324	0.247	0.312	0.244	0.334	0.297	

**Table 6 behavsci-15-00582-t006:** F-square values.

Construct	F-Square
CC → PSI	0.023
COL → PSI	0.029
FE → PI	0.064
HOM → PSI	0.027
PSI → FE	0.043
PSI → PI	0.023
EC → PSI	0.022

**Table 7 behavsci-15-00582-t007:** Path coefficients and hypothesis testing.

Hypothesis	Paths	Path Factor	Sample Mean	STDEV	*t*-Values	*p*-Values	Conclusion
H1	COL → PSI	0.165	0.169	0.044	3.763	0.000 ***	Support
H2	EC → PSI	0.143	0.145	0.043	3.305	0.001 ***	Support
H3	CC → PSI	0.149	0.150	0.041	3.633	0.000 ***	Support
H4	HOM → PSI	0.158	0.159	0.042	3.715	0.000 ***	Support
H5	PSI → FE	0.202	0.205	0.041	4.899	0.000 ***	Support
H6	PSI → PI	0.147	0.148	0.044	3.363	0.001 ***	Support
H7	FE → PI	0.245	0.249	0.040	6.058	0.000 ***	Support

Note: *** *p* < 0.001.

**Table 8 behavsci-15-00582-t008:** Mediation effect test.

Hypothesis	Paths	Path Factor	Sample Mean	STDEV	t Values	*p* Values	Conclusion
H8	COL → PSI → FE → PI	0.008	0.009	0.003	2.526	0.012 *	Support
H9	EC → PSI → FE → PI	0.007	0.007	0.003	2.263	0.024 *	Support
H10	CC → PSI → FE → PI	0.007	0.008	0.003	2.551	0.011 *	Support
H11	HOM → PSI → FE → PI	0.008	0.008	0.003	2.647	0.008 **	Support

Note: * *p* < 0.05, ** *p* < 0.01.

## Data Availability

The raw data supporting the conclusions of this article will be made available by the authors on request.

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
