# Peer review of "How Collectivism and Virtual Idol Characteristics Influence Purchase Intentions: A Dual-Mediation Model of Parasocial Interaction and Flow Experience"

_behavsci, 2025, doi:10.3390/bs15050582_

Round 1

Reviewer 1 Report

Comments and Suggestions for Authors

The paper addresses an interesting and current topic with an emphasis on digital marketing and consumer behavior, focusing on the role of virtual idols within the context of collectivist culture. Despite the clearly significant research area, I suggest the following improvements to the paper:

  1. The paper should clearly state in which country the survey was conducted (even though it is apparent that it was in China), during which time, and how the sample of respondents was selected.
  2. PSI is a key concept in the paper, but a more in-depth elaboration is recommended—for example, distinguishing between passive content viewing and interactive engagement with virtual idols.
  3. As the research was conducted exclusively in China, it is recommended to consider a comparison with individualistic cultures to further test the universality of the proposed model.
  4. Although the inclusion of a "screening question" is mentioned, the actual example of the questionnaire is not provided. It is recommended to include it in an appendix for transparency.
  5. In several parts of the text, there are sentences that come across as telegraphic (e.g., "PSI and flow experience as dual mediators"). It is recommended to revise the language for improved fluency and cohesion.
Comments on the Quality of English Language

In several parts of the text, there are sentences that come across as telegraphic (e.g., "PSI and flow experience as dual mediators"). It is recommended to revise the language for improved fluency and cohesion.

Author Response

Comments 1: The paper should clearly state in which country the survey was conducted (even though it is apparent that it was in China), during which time, and how the sample of respondents was selected.

Response 1: Thank you for pointing this out. We agree with this comment. Therefore, we have added detailed information about the sample and data collection procedures in Section 3.2, including the country, timeframe, and recruitment method(pp. 8-9, lines 360–369).

Comments 2: PSI is a key concept in the paper, but a more in-depth elaboration is recommended—for example, distinguishing between passive content viewing and interactive engagement with virtual idols.

Response 2: Thank you for pointing this out. We agree with this comment. Therefore, we have expanded the explanation of parasocial interaction (PSI), clarifying its connection to interactive engagement with virtual idols and explicitly distinguishing it from passive content viewing. As PSI refers to perceived interaction, it inherently involves a sense of active engagement rather than passive content viewing(pp. 5-6, lines 202–232).

Comments 3: As the research was conducted exclusively in China, it is recommended to consider a comparison with individualistic cultures to further test the universality of the proposed model.

Response 3: Thank you for pointing this out. We agree with this comment. Therefore, in response to the recommendation to compare collectivist and individualist cultures, we have added relevant theoretical discussion to the “Discussion” section (p. 14, lines 474–484). Since the current study is based solely on data from China, we also address this limitation in the “Limitations and Future Research” section, where we suggest cross-cultural empirical comparisons in future studies(p. 16, lines 591-594).

Comments 4: Although the inclusion of a "screening question" is mentioned, the actual example of the questionnaire is not provided. It is recommended to include it in an appendix for transparency.

Response 4: Thank you for pointing this out. We agree with this comment. Therefore, the descriptive statistics of the questionnaire sample have been provided as a screenshot and placed in Appendix B, as the questionnaire was originally administered in Chinese. The full set of survey items has been translated into English and presented in table format. These items are included in Appendix A for reference.

Comments 5: In several parts of the text, there are sentences that come across as telegraphic (e.g., "PSI and flow experience as dual mediators"). It is recommended to revise the language for improved fluency and cohesion.

Response 5: Thank you for pointing this out. We agree with this comment. Therefore, we have reviewed and revised the manuscript throughout to improve fluency, address overly telegraphic language, and ensure better cohesion in academic expression.

Reviewer 2 Report

Comments and Suggestions for Authors

The paper presents a timely and relevant examination of how collectivism and virtual idol characteristics influence consumer purchase intentions through parasocial interaction (PSI) and flow experience. While the topic is theoretically grounded and empirically supported, several areas require improvement to enhance the paper’s clarity, coherence, and contribution.

The introduction, although rich in context, is overly long and includes redundant details, such as extended examples of celebrity endorsement controversies. The transition from general marketing to virtual idol marketing could be more sharply articulated, and the novelty of the research is overstated, as prior studies have explored similar constructs in East Asian contexts. I recommend moving Table 1 to the beginning of the Literature Review section. The introduction should build an argument and set up the research questions smoothly; inserting a table can interrupt this flow. References could be updated. More references on virtual influencers/AI influencers in 2024-25 could be added (e.g., Wang, Y., Tang, Z., Wang, W., Zhao, D., He, D., & Lu, Y. (2025). A systematic literature review of virtual idol from the perspective of the business role ecosystem. Internet Research https://doi.org/10.1108/INTR-06-2024-0938; Feng, Y., Chen, H., & Xie, Q. (2024). AI influencers in advertising: the role of AI influencer-related attributes in shaping consumer attitudes, consumer trust, and perceived influencer–product fit. Journal of Interactive Advertising24(1), 26-47.) The authors could more clearly state the research gap and how this study fills it.

The literature review is overly descriptive and lacks synthesis. Although the segmentation of virtual idol characteristics into external, content, and homophily is a strength, the discussion overlaps at times and would benefit from clearer theoretical distinctions. The hypothesis development sections are repetitive, and many statements reiterate the same logic without critical engagement or contrasting viewpoints.

The theoretical model is conceptually sound but would benefit from earlier placement of the visual diagram and a deeper integration of competing or complementary frameworks. The heavy reliance on Social Identity Theory and Flow Theory may limit the broader applicability of the findings, especially since the potential role of moderators (e.g., gender or digital familiarity) is not explored.

Methodologically, the use of WeChat for data collection and monetary incentives raises questions about sampling bias and generalizability. The screening procedure for identifying knowledgeable participants is mentioned but under-explained, and the representativeness of the sample is skewed toward younger, middle-income individuals. While the results are statistically robust, the reporting is dense and overly technical in places; several measurement and validity tables could be moved to an appendix to improve readability. Mediation results are presented clearly, but further interpretation and visualization (e.g., with path diagrams) would help readers unfamiliar with structural equation modeling.

The discussion section tends to restate results rather than offering new theoretical insights or critical reflections. There is limited engagement with alternative explanations or limitations of PSI and flow as psychological constructs. The theoretical and managerial implications are not clearly separated, and the latter tend to be generic, lacking grounding in specific data points or real-world examples. The implications section could be more actionable if linked directly to findings and accompanied by concrete brand case studies. Lastly, while the limitations section appropriately notes the cross-sectional design, it does not address the lack of cultural comparison or the risk of social desirability bias. Future research suggestions remain general and could be expanded to include longitudinal designs or experimental approaches. Overall, while the paper offers valuable contributions to the growing literature on virtual idols and consumer behavior in collectivist contexts, it would benefit from tighter writing, greater synthesis, and a more critical analytical lens.

Author Response

Comments 1: The introduction, although rich in context, is overly long and includes redundant details, such as extended examples of celebrity endorsement controversies. The transition from general marketing to virtual idol marketing could be more sharply articulated, and the novelty of the research is overstated, as prior studies have explored similar constructs in East Asian contexts. I recommend moving Table 1 to the beginning of the Literature Review section. The introduction should build an argument and set up the research questions smoothly; inserting a table can interrupt this flow. References could be updated. More references on virtual influencers/AI influencers in 2024-25 could be added (e.g., Wang, Y., Tang, Z., Wang, W., Zhao, D., He, D., & Lu, Y. (2025). A systematic literature review of virtual idol from the perspective of the business role ecosystem. Internet Research https://doi.org/10.1108/INTR-06-2024-0938; Feng, Y., Chen, H., & Xie, Q. (2024). AI influencers in advertising: the role of AI influencer-related attributes in shaping consumer attitudes, consumer trust, and perceived influencer–product fit. Journal of Interactive Advertising24(1), 26-47.) The authors could more clearly state the research gap and how this study fills it.

Response 1: Thank you for pointing this out. We agree with this comment. Therefore,

  1. The section discussing celebrity endorsement controversies has been removed to avoid redundancy and improve focus.
  2. We have improved the logical flow by clarifying the transition from general marketing to virtual idol marketing and more directly highlighting the relevance of the research context (pp. 1-2, lines 39-50).
  3. In the contribution section, we revised our language to avoid overstating novelty in the East Asian context. Rather than emphasizing regional originality, we now focus on conceptual and methodological contributions (p. 3, lines 121-134).
  4. Following the reviewer’s suggestion, Table 1 has been relocated to the beginning of the Literature Review section to maintain the coherence of the introduction (p. 4, lines 143-150).
  5. We have updated the reference list to include more recent studies published in 2024 and 2025, such as Wang et al. (2025) and Feng et al. (2024), to strengthen the foundation and currency of the literature(p. 2, lines 51-62).

Comments 2: The literature review is overly descriptive and lacks synthesis. Although the segmentation of virtual idol characteristics into external, content, and homophily is a strength, the discussion overlaps at times and would benefit from clearer theoretical distinctions. The hypothesis development sections are repetitive, and many statements reiterate the same logic without critical engagement or contrasting viewpoints.

The theoretical model is conceptually sound but would benefit from earlier placement of the visual diagram and a deeper integration of competing or complementary frameworks. The heavy reliance on Social Identity Theory and Flow Theory may limit the broader applicability of the findings, especially since the potential role of moderators (e.g., gender or digital familiarity) is not explored.

Response 2: Thank you for pointing this out. We agree with this comment. Therefore,

  1. We have removed overlapping and repetitive content between the theoretical background and the hypothesis section to improve clarity.
  2. We have added more critical discussion and included contrasting perspectives to strengthen the development of hypotheses(p. 6, lines 247-250; p. 6, lines 262-264; p. 8, lines 331-334).
  3. Figure 1 has been moved to follow the theoretical framework for better logical flow(p. 6, lines 232-233).
  4. We have also noted the reliance on Social Identity Theory and Flow Theory, and acknowledged the absence of moderator variables. These limitations have been discussed in the “Limitations and Future Research” section(p. 16, lines 582-589).

Comments 3: Methodologically, the use of WeChat for data collection and monetary incentives raises questions about sampling bias and generalizability. The screening procedure for identifying knowledgeable participants is mentioned but under-explained, and the representativeness of the sample is skewed toward younger, middle-income individuals. While the results are statistically robust, the reporting is dense and overly technical in places; several measurement and validity tables could be moved to an appendix to improve readability. Mediation results are presented clearly, but further interpretation and visualization (e.g., with path diagrams) would help readers unfamiliar with structural equation modeling.

Response 3: Thank you for pointing this out. We agree with this comment. Therefore,

  1. We have added further explanation of the screening procedure used to identify knowledgeable participants, clarifying the criteria applied during the data collection process(pp. 8-9, lines 359-368).
  2. Regarding concerns about sampling bias and generalizability, we note that the sample is primarily composed of younger, middle-income individuals. Rather than a limitation, this reflects the actual demographic of virtual idol consumers in China, who are typically digital natives with moderate purchasing power.
  3. To improve readability, we have moved the full list of measurement items to the appendix A, thereby reducing the density of information in the main text.
  4. In the mediation analysis section, we have revised the explanation of the results shown in Table 8 to make the interpretation clearer and more accessible for readers who may be less familiar with structural equation modeling(p. 13, lines 444-448).

Comments 4: The discussion section tends to restate results rather than offering new theoretical insights or critical reflections. There is limited engagement with alternative explanations or limitations of PSI and flow as psychological constructs. The theoretical and managerial implications are not clearly separated, and the latter tend to be generic, lacking grounding in specific data points or real-world examples. The implications section could be more actionable if linked directly to findings and accompanied by concrete brand case studies. Lastly, while the limitations section appropriately notes the cross-sectional design, it does not address the lack of cultural comparison or the risk of social desirability bias. Future research suggestions remain general and could be expanded to include longitudinal designs or experimental approaches. Overall, while the paper offers valuable contributions to the growing literature on virtual idols and consumer behavior in collectivist contexts, it would benefit from tighter writing, greater synthesis, and a more critical analytical lens.

Response 4: Thank you for pointing this out. We agree with this comment. Therefore,

  1. We have expanded the discussion to include new theoretical insights and interpretations beyond the descriptive reporting(p. 14, lines 499-506; p. 15, lines 512-517; p. 15, lines 518-530).
  2. We have addressed the conceptual limitations of PSI and flow, acknowledging their subjectivity and theoretical boundaries(p. 16, lines 582-586).
  3. We have revised the implications section to clearly distinguish between theoretical and managerial implications(pp. 15-16, lines 532-576).
  4. Following the reviewer’s recommendation, we have also enhanced the managerial implications by incorporating concrete brand examples to demonstrate how virtual idols are applied in marketing strategies(p. 15, lines 532-558).
  5. We have revised the “Limitations and Future Research” section to include additional considerations, such as the absence of cross-cultural comparison, the potential for social desirability bias, and suggestions for future studies using longitudinal or experimental designs(p. 16, lines 578-594).
